# Improving the Suitability of Vaccine Design for Immunisation Programmes and Enhancing Vaccine Policy Quality Through User Research

**DOI:** 10.3390/vaccines13101075

**Published:** 2025-10-21

**Authors:** Stefano Malvolti, Melissa Malhame, Adam Soble, Carsten Mantel, Melissa Ko, Lorena Perrin, Tiziana Scarna, Marion Menozzi-Arnaud, Jean-Pierre Amorij

**Affiliations:** 1MMGH Consulting GmbH, 8049 Zurich, Switzerland; malhamem@mmglobalhealth.org (M.M.); soblea@mmglobalhealth.org (A.S.); kom@mmglobalhealth.org (M.K.); 2Impact Hub, 8005 Zurich, Switzerland; lorena.perrin@impacthub.net; 3Gavi, The Vaccine Alliance, 1218 Geneva, Switzerland; tiziana.scarna@gmail.com (T.S.); mmenozziarnaud@gavi.org (M.M.-A.); 4UNICEF Supply Division, 2150 Copenhagen, Denmark; jamorij@unicef.org

**Keywords:** IA2030, vaccines, use cases, user research, design approach, implementation strategy, product development

## Abstract

Background: The achievement of the goals of the Immunization Agenda 2030 (IA2030) requires vaccines to be developed and implemented that meet the needs and requirements of the final users: vaccinators and vaccinees. A detailed and shared understanding of these needs should inform policy and programme guidance directing stakeholders’ efforts and investments. Currently, relevant guidance documents only partially capture vaccine users’ perspectives. Method: To help overcome this gap, we propose an operational research method grounded in the principles of the design approach that systematically maps and integrates user perspectives in vaccine development, policy, and implementation decisions. Results: The method, named the seven Ws, guides researchers through a three-step process. First, it clarifies the contribution of a vaccine to solving a public health problem—the solution–problem fit. Second, it maps potential implementation strategies for the vaccine in different settings. Lastly, it describes the relevant vaccine’s use cases across the implementation strategies, elucidating the user requirements for the vaccine to be successfully implemented—the solution–provider and solution–user fits. Conclusions: By explicitly pursuing these three fits, policymakers, vaccine developers, and programme managers will be able to better contribute towards the achievement of the IA2030 goals. This framework is intended as a conceptual contribution rather than an empirical validation study.

## 1. Introduction

For vaccines to deliver their full value and contribute to the achievement of the goal of the Immunization Agenda 2030 (IA2030) of “a world where everyone, everywhere, at every age, fully benefits from vaccines for good health and well-being [1]”, they must reach their target population while retaining their full potency. Meeting IA2030′s goal is dependent on a combination of interrelated factors: some programme-related (convenience of the administration, duration of the vaccination session, location and frequency of administration, training requirements, etc.), some product-related (physical properties of antigens and possible adjuvants, formulation, number of doses, side effects, etc.), and some user-related (awareness, acceptability, ease of use, etc.).

Many of these factors result from product design choices made during the early phases of vaccine development, up to 10 years before the vaccine reaches first marketing authorisation, and often many years before programmatic and policy decisions are taken. In those phases, factors related to product development and manufacturing exert maximum influence on final product attributes, in conjunction with commercial and financial aspects [2]. Whether vaccines are suitable from a user, programme, affordability, geographical, and cultural standpoint often receives more limited attention. Furthermore, since vaccines are developed almost exclusively by for-profit entities, the focus is often directed towards the needs of more profitable high-income markets. This results in choices of product attributes that may be suboptimal for other geographies, particularly in Low- and Middle-Income Countries (L&MICs). Multiple examples of this problem exist:A Rotavirus vaccine developed with a large packaging volume, putting undue strain on the cold chain, and not being equipped with a vaccine vial monitor, prohibiting the monitoring of temperature excursions [3].A Pneumococcal Conjugate Vaccine (PCV) developed selecting the serogroups more prevalent in North America and Europe [4,5].Hexavalent and pentavalent vaccines developed with an acellular pertussis component that reduces potential reactogenicity while sacrificing a stronger or broader immune response [6,7].Some of the vaccines under development for Shigellosis did not include young children in low-income countries early enough in clinical trials, potentially affecting the suitability of the vaccine for the population most in need [8].

Once defined, these choices are irreversible, since they define the design of clinical trials and the specifics of the manufacturing process that constitute the main elements of the regulatory dossier. While opportunities exist for later adjustments to fit specific users’ needs, these changes are expensive, time-consuming, and may result in the delayed introduction of life-saving vaccines. To achieve the IA2030 goals, vaccines need to be suitable for all settings, particularly those with the highest needs. This will only be possible if vaccine product attributes and development decisions more comprehensively consider the needs of users and programmes in all settings.

### 1.1. The Current Global Guidance and Its Focus

Currently, guidance to vaccine developers on the desired features of vaccines is provided by major public health entities as part of various publications. Among those, the ones that provide specific input on product attributes are (as summarised in Table 1):The Preferred Product Characteristics (PPCs), which are published by the World Health Organization (WHO) for all critical vaccines in early stage clinical development and define the attributes that optimise vaccine use and contribute to meeting global public health needs [9]. While focused on L&MICs, the PPCs are an important reference for vaccine developers.The Target Product Profiles (TPPs) define product attributes such as indication, target population, dosing regimen, duration of protection, route of administration, safety, and efficacy [9]. In the case of TPPs developed by the WHO, particular focus is devoted to requirements to be fulfilled by products seeking WHO policy recommendation and prequalification (PQ). Other organisations besides the WHO also develop TPPs, such as CEPI (the Coalition for Epidemic Preparedness and Innovation) [10] or the Center for Biologics Evaluation and Research (CBER) [11], albeit with more targeted and focused objectives.Evidence Considerations for Vaccine Policy Development (ECVP), developed by the WHO, aims to provide early guidance on the evidence likely required to support WHO policy recommendations [12]. The first ECVP was published for novel TB vaccines [13].The technical document guiding the Assessment of the Programmatic Suitability of Vaccine Candidates for WHO Prequalification (PSPQ) specifies the programmatic requirements for use in L&MICs (mandatory, critical, and preferred) necessary to achieve WHO PQ [14].

Except for the ECVP, which expands on the programmatic and contextual aspects of vaccine use, inclusive of the social and cultural context of immunisation as well as of the interaction with other health and non-health programmes, these guidance documents capture primarily technical product aspects (clinical, safety, dosing, presentation, etc.). Users’ perspectives, namely those of the health workers, the vaccinees, and the caregivers (where applicable), are subject to a less systematic assessment. They emerge when the vaccine is at the end of the development process, as in the cases of the PCV [15] and Malaria vaccine [16]. Such validation occurs much less frequently ex ante, as one of the key factors guiding programme and product design choices.

With this paper, we propose expanding the base of evidence informing the definition of product guidance documents and of vaccine development decisions, including more systematically examining users’ perspectives. The proposed approach aims at complementing existing guidance in informing vaccine development, policy choices, and programme design. Putting people and their needs at the core of health services and of the medical products they will use is critical to produce better outcomes and increase people’s ownership and trust in the health system.

### 1.2. The Need for a User-Centric Focus in Vaccine Development, Policy, and Programme Design

The systematic adoption of a design approach can respond to the need to increase focus on users’ needs from early product development to final programme design decisions. The core of design work is studying users’ practices and attitudes towards products and services to generate insights that can lead to improvements, which, in our case, would be better vaccines and better immunisation services. It is all about moving from the current state (“as is”) to achieve the goal of a desired future improved state (“to be”). Adopting a design approach that focuses on the users is ultimately about generating a “fit” across several different elements [17]. This approach does not focus solely on the “technical” aspects of the health problem but rather on the broader achievement of three “fits” (Figure 1):
Solution–problem fit: the solution should provide the right fit for the health problem—i.e., the vaccine attributes should address the disease control or elimination goals (e.g., a pandemic vaccine capable of blocking transmission).Solution–user fit: the solution should fit with the users’ capabilities, preferences, and needs—i.e., the vaccine attributes should be acceptable for vaccinees and caregivers (e.g., a vaccine that meets specific community requirements).Solution–provider fit: the solution should fit with the operational processes of the provider(s)—i.e., the vaccine attributes should be manageable by the immunisation infrastructure and health workers (e.g., a vaccine that does not require Ultra Cold Chain—UCC—in areas where this is not available).

**Figure 1 vaccines-13-01075-f001:**
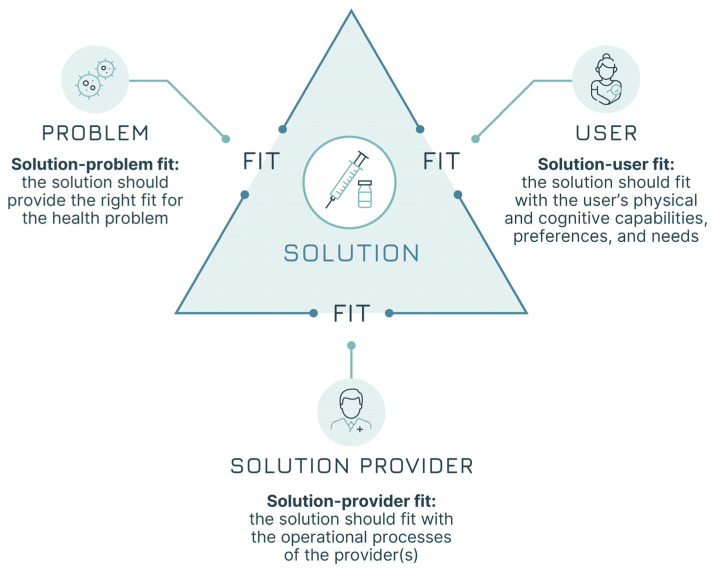
The three fits (adapted from Leurs et al. [17]).

The approach spans the entire life cycle of a vaccine, starting from its design and development and covering all aspects of service delivery. During the product development phase, early insights into vaccine implementation strategies and users’ needs can help anticipate policy recommendations and guidance on the potential use of a vaccine. An early understanding of the use cases should identify evidence gaps for policy generation to ensure that the appropriate data are available by the time policies will need to be formulated. Finally, the approach of the three fits should also guide the design of clinical trials and the prioritisation of product-specific attributes that would be especially valuable for the users. At the other end of the life cycle, once vaccines have been introduced in national immunisation programmes (NIPs), a better understanding of users’ perspectives and needs could highlight potential areas for improvement of second-generation products and help to conceptualise interventions for increased use and coverage.

Multiple trade-offs exist at each step of the product development and life cycle where choices need to be made on product attributes. An example concerns the trade-off between single and multi-dose presentations. The former is more convenient for health workers and creates almost no wastage but is more costly and requires more cold chain space; the latter is less expensive and less demanding on the cold chain but can be the cause of open-container wastage and relies on the ability of health workers to draw the correct dosage from the vial. Different resolutions of such trade-offs impact the time and cost to develop and implement the vaccine, as well as its overall financial, economic, and health impact. Attention to users’ perspectives—in this case those of the health workers and programme managers—can prove critical in ensuring that those trade-offs are resolved in a balanced way, increasing the likelihood of achieving maximum coverage and health impact.

In the service delivery space, UNICEF has started moving in this direction by developing a general framework [18] and presenting successful examples, such as the example of Malaysia, where the adoption of a design approach that put users at the centre helped build local capacity and design more child-centred services. More specific to immunisation, two recent projects focused on adjusting the design of immunisation programmes at the local level to help increase COVID-19 vaccine coverage in Malawi and to define context-specific interventions as part of urban immunisation services in Madagascar [19].

With the goal of facilitating more widespread adoption of a design approach to inform vaccine development and implementation, we propose a set of definitions and a logical framework to investigate users’ requirements and solution fits.

The concepts laid out in this paper are addressed primarily to agencies and donors considering extending their support to vaccine development programmes, policymakers charged with defining vaccine implementation strategies and policies, vaccine developers as they design their clinical development programmes and decide on vaccine product characteristics, and immunisation professionals guiding the implementation of vaccine programmes.

This article proposes a methodological framework to systematically map and integrate user perspectives into vaccine development, policy, and implementation. Its contribution is primarily conceptual and does not constitute an empirical validation. The framework is not intended to replace established approaches but to complement them by placing greater emphasis on users’ needs and requirements. While the identification of user needs and the definition of relevant use cases represent key dimensions of product design, there are other critical components. Rigorous assessments across additional domains—including safety, regulatory, manufacturing, and financing—are required to determine the viability of the design recommendations generated through this approach.

To demonstrate its practical application, five case studies are presented that illustrate the implementation of the framework and the types of outputs it generates.

## 2. The Seven Ws Method and Case Studies

With the aim of facilitating consideration of users’ perspectives in vaccine development, immunisation policy design and programme decisions, we have developed a mixed-methods research approach [20]. The method builds on fundamental features of the design approach and aims at achieving granular clarity on the solution—the vaccine—and on the users—vaccinees and vaccinators. It does so by keeping in mind the critical role of policymakers (at local, regional, and global levels) and other relevant stakeholders (e.g., government officials, donors, procurement agents, supply chain professionals, etc.). The research approach examines the factors influencing how a vaccine is or could be used across different implementation strategies: its use cases.

The original definition of use case evolved in software development in 1987 and has become an integral part of the Unified Modelling Language (UML), the standard way of visualising the design of a system [21]. In a UML context, use cases are intended as “all the ways of using a system to achieve a particular goal for a particular user”. We adapted this definition to capture the interaction between a user, a provider, and a health product or service [22] in a healthcare context; therefore, we define a use case as “*a specific situation where a health product or a service is or can be used to achieve a defined health goal*”.

Our approach foresees three steps aimed at testing the three fits to identify gaps and areas for improvement (Figure 2):Step 1: The generic Solution-problem fit is tested to clarify the way in which the vaccine is going to be used. In this first step, the first two Ws are examined: the WHY (the public health problem the vaccine seeks to address) and the WHAT (the vaccine researched).Step 2: All three fits (Solution-problem, Solution-provider, and Solution-user) are assessed across the programmatic and policy contexts in which the vaccine is likely to be used. This leads to the identification of the implementation strategies most relevant for the vaccine. In this step, the remaining 5 Ws are addressed, with a focus on the user groups: the WHO (the health programme responsible for delivering the vaccine—not to be confused with the acronym used for the World Health Organization), the WHOM (the populations targeted), the WHEN (the timing of administration and the epidemiological context), and the WHERE and the WITHIN (the geographical and health system context).Step 3: The three fits are assessed a second time, focusing, within each implementation strategy, on the specific field situations in which the vaccine is used, e.g., the user–product interaction. This step leads to the identification of vaccine use cases. The three Ws relevant at the user level are revisited, shifting the focus from groups and generic settings to individual users and vaccination sites, i.e., redefining the who (as the vaccinator), the whom (as the vaccinee) and the where (as the location of the administration).

**Figure 2 vaccines-13-01075-f002:**
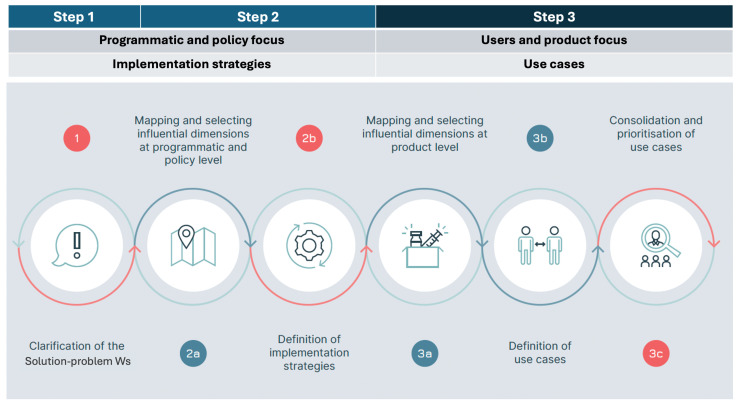
The three steps of the seven Ws method.

The goal of this approach is to identify all potential “uses” of the vaccine. Accordingly, possible upstream limitations such as manufacturing, financial, policy, regulatory, procurement, and supply chain constraints, are deliberately excluded from this approach. Their consideration at this stage could lead to the premature elimination of some of the use cases with potential to address the public health problem in question (the WHY). The systematic identification of all possible use cases (the “to be” state), including less apparent or straightforward ones, is one central contribution of this approach.

### 2.1. Step One—The General Assessment of the Solution–Problem Fit

The foundational step of our approach is the exploration of the Solution–problem fit (Figure 3). This entails achieving clarity on two components:The problem (what we will call the WHY)—the public health problem, defined by the epidemiology of the disease, that the vaccine seeks to reduce, with all the variations across different demographic, economic, and socio-cultural contexts. (i.e., To what end or, in other words, WHY would the solution be used?)The solution (what we will call the a)—the vaccine used or the candidate vaccine being developed to address the health problem. The vaccine’s attributes influence who can administer the vaccine (e.g., only trained health workers can practice intramuscular injections instead of community health workers, who can only administer oral vaccines), who can receive them (based on the indication), and where the vaccine can be administered (e.g., if a cold chain is required, there may be limitations in the locations that allow delivery of such a vaccine). Description of the vaccine attributes should adopt a forward-looking perspective focused on the “to-be” and be capable of capturing the full potential of the vaccine based on desirable attributes that may have yet to be confirmed.

**Figure 3 vaccines-13-01075-f003:**
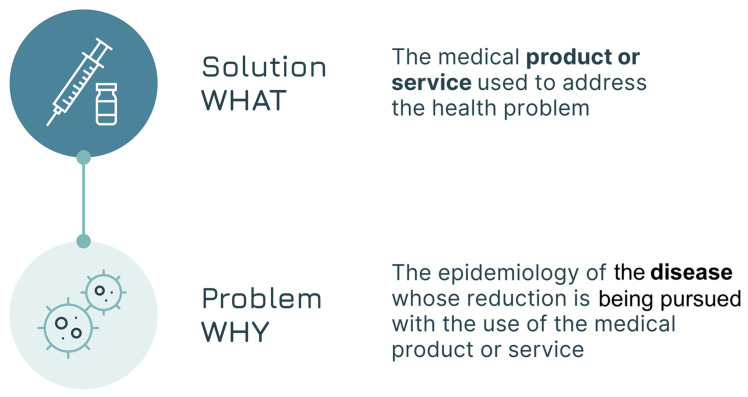
The Solution–problem fit.

While analysis of the Solution–problem fit may seem straightforward, it is important to ensure both comprehensiveness and specificity. On the problem side, it is critical that different disease characteristics (e.g., epidemic-prone vs. endemic disease) and the different health goals pursued (e.g., prevention of disease or of infection or transmission) are clearly described. On the solution side, all relevant vaccine attributes need to be well-understood to ensure that the test of the fit is sufficiently specific (Box 1, Case Study 1). This first step of the analysis can already elicit valuable insights that help with focusing development and policy choices.

Box 1Case Study 1: the definition of the WHY for a Pan-Sarbecovirus (e.g., SARS-related/SARS-like betacoronavirus) vaccine to be used in L&MICs. Context: As part of a project to define implementation strategies and high-level use cases for a Pan-Sarbecovirus vaccine, an expert group comprising regional and global stakeholders identified the public health problems that such a vaccine could address.
Example: the public health problem(s) that could be addressed by a Pan-Sarbecovirus vaccine with specific characteristics relevant for L&MICs were defined. The objectives (WHY) for the use of the vaccine were identified both in the context of an outbreak of a novel Sarbecovirus (SARS-X outbreak) and during times of ongoing circulation of SARS-CoV-2 variants.
During an outbreak of SARS-X, three potential goals were defined:
preventing severe disease (and thereby also mortality) caused by SARS-Xminimising the impact on health and economic systems in any part of the world while maintaining an acceptable benefit/risk ratioIdeally reducing infection and transmissionDuring the ongoing circulation of SARS-CoV-2 variants, one primary goal was defined:
providing improved protection against symptomatic infection caused by SARS-CoV-2 compared to other available licenced vaccines (with other desirable attributes being equivalent safety, improved durability, improved protection against long COVID, and lower cost)

#### Methodology and Tools for the Assessment of the Solution–Problem Fit

The active involvement of researchers from high-burden countries—whether individual experts or research institutions—is recommended. Their participation ensures contextual understanding of local specificities and improved access to stakeholders and data sources, as well as a more resource-efficient process.

To coordinate the research, a steering group should be established comprising relevant country, regional, and global stakeholders. This group will provide methodological oversight and will validate the outputs. In parallel, a working group should be assembled, including representatives from the research entities involved at the country and global level. This group will consolidate the country’s perspectives and will be responsible for research outputs. Meetings of these groups should be conducted virtually. At the outset, Terms of Reference and a meeting calendar for both groups will need to be defined, as well as a RACI for the latter. Appropriate videoconferencing and data-sharing platforms will need to be established.

The working group will perform the following activities:A desk review of the published and grey literature to generate a preliminary overview of the problem–solution fit. The review should cover the following:○The vaccine characteristics (the WHAT)—if the product already exists or is in late-stage development, the TPP will serve as the primary reference. If the research targets a vaccine in the earlier stages of development, the PPC or an equivalent document can be used. If no such document exists, a provisional profile could be developed in consultation with subject-matter experts.○The disease context (the WHY)—the review should compile evidence on epidemiology, disease burden, and available diagnostic, treatment, and prevention options. The focus will be on high-burden countries and disadvantaged or at-risk populations.A Rapid Insight Validation should be conducted through interviews or a small focus group (6–8 participants). These activities will be conducted virtually.

Outputs:A Solution–problem briefing including a summary description of the WHY and the WHAT, an overview of the risks and assumptions, and a mapping of the quality of the evidence and hypotheses regarding country archetypes to be validated in Step Two.Documentation of the desk review approach, selection and exclusion criteria, and selected sources of information.

### 2.2. Step Two—The Assessment of the Three Fits at the Programmatic and Policy Level (Definition of the Implementation Strategies)

Once the Solution–problem elements are identified, a policy and programmatic lens is adopted to allow for the assessment of the specific context in which the vaccine will be used. The goal is the description of likely implementation strategies. With the WHY and WHAT in mind, as defined in step one, the analysis continues by looking at the main contextual factors that influence the use of the vaccine:The geographical setting (country, region, or other subnational context) in which, based on epidemiological characteristics, the vaccine will be used (the WHERE) to provide a solution to the health problem. This includes both currently endemic areas as well as areas with populations at risk of infection.The “health systems” of the geographies identified, with their constraints, norms, and resources which define the context of vaccine use as part of a broader set of health services delivered to the population (the WITHIN).The timing and frequency of the vaccine administration and, if applicable, its temporally linked co-administration with other health products (the WHEN).

Once those contextual factors are clarified, the analysis concentrates on the user groups, intended as the population and programmes involved in the use of the vaccine (Box 2, Case Study 2):The health programme that delivers the vaccine in each country and its relationship with other programmes in and outside the healthcare space whose contribution may be required or desirable (the WHO).The population or populations targeted (the WHOM). For this “W”, the adoption of an equity-centred approach calls for special attention to be dedicated to the needs and perspectives of hard-to-reach populations (e.g., nomadic groups, undocumented migrants, etc.) and underrepresented communities and groups.

Box 2Case Study 2: the definition of the WHOM for a vaccine against pulmonary tuberculosis (TB) at the policy–programmatic level: the population [23].Context: As part of a project to assess the market size, implementation strategies, and use cases for a tuberculosis vaccine for adolescents and adults, a desk review and interviews with 27 representatives across 10 countries were conducted to inform the prioritisation of key target populations. This prioritisation was carried out by a multidisciplinary steering group of experts at the country, regional, and global level during a virtual meeting.
Example: A generic new TB vaccine can target different populations depending on its characteristics and indications. These can be identified by looking at

-Age groups;TB infection status;HIV status.
Other characteristics of the targeted populations can provide opportunities for accessing particular delivery channels other than the traditional ones used for immunisation:
-Pregnancy status;Presence of co-morbidities;Nutritional status (e.g., undernourishment, obesity, Type 2 diabetes);Substance use disorder status (e.g., alcohol, drugs);Living in high congregated settings (e.g., mines, prisons, schools, refugee camps, etc.).

The elements analysed in this step are illustrated in Figure 4.

Analysing the seven Ws at the programmatic and policy levels allows for the identification of the vaccine implementation strategies that can be employed and that are most likely to maximise the Solution–user, the Solution–provider, and ultimately the Solution–problem fits (marked in red in Figure 4). An implementation strategy is defined as the “*methods or techniques used to enhance the adoption, implementation, and sustainability of a clinical program or practice*” [24], i.e., the appropriate combination of policies, personnel, resources (financial and technical), operational approaches, and behavioural aspects.

Once the potential implementation strategies are defined, a further review of the contextual dimensions, as described in the WHERE and WITHIN, is performed to validate their relevance. Some implementation strategies may only be relevant in specific countries while others may be common across different geographies.

This analysis allows for the definition of country archetypes intended as a “*group of countries sharing common characteristics concerning the use of a specific vaccine or in the design of a specific health program*”. The implementation strategies shared across the archetypes and with maximum expected impact are prioritised (Figure 5) and taken forward to the third step of our approach.

By combining the seven Ws, we ultimately obtain a narrative description for each of the “implementation strategies”. The following is an example that refers to a potential implementation strategy for a measles–rubella vaccine:

“A measles-rubella combination vaccine (the WHAT) is administered during an outbreak response (the WHEN) by the NIP (the WHO) to a population aged 9 months to 15 years (the WHOM) in countries in Central Asia (the WHERE), with decentralised health systems and high vaccine hesitancy (the WITHIN), to achieve the goal of eliminating measles (the WHY).”

#### Methodology and Tools for the Programmatic and Policy Analysis

A mixed-methods programmatic and policy analysis is conducted to capture the key features of national health systems, policies, and relevant cultural or social factors (the WHERE and the WITHIN); the view points and needs of the targeted population (the WHOM) with particular emphasis on marginalised and at-risk groups; the different times in which the vaccine can be administered (the WHEN); and the perspectives of health workers (the WHO) across different levels of the health system.

In selected archetype countries, local research teams should conduct:A document and policy scan to analyse EPI plans, budgets, guidelines, role of the private sector, and integration with primary healthcare.Stakeholder mapping to identify key actors involved in the decisions and in the use of the vaccine, assessing both their interests and influence with special attention given to the users (vaccinees and vaccinators). The findings of this mapping should inform the sampling criteria for conducting surveys and interviews.A user and stakeholder survey (≈20–30 respondents per country with appropriate sampling approaches) using appropriate tools (e.g., free-of charge Kobo/ODK) to capture health system status, delivery channels, health worker competencies, user requirements and perceptions, and potential implementation barriers.Qualitative interviews (≈5–8 per country) with policymakers, regulators, health workers, programme managers, and community representatives to validate findings from prior activities. Interviews will be recorded and coded; an inter-coder agreement of ≥0.7 should be achieved to ensure reliability of cross-country insights.

Based on the output of the country activities, the working group should perform the following:Definition of country archetypes: Countries will be clustered according to key features and constraints (e.g., outreach-heavy vs. facility-heavy; high hesitancy vs. high demand).Drafting of implementation strategies: For each archetype, short narrative descriptions of the implementation strategies will be developed by combining the seven Ws, identifying enabling conditions and barriers. The draft document will serve as a pre-read for the validation workshop.

An implementation strategies workshop should be convened with representatives from key stakeholder groups (ideally with ≥40% participation from high-burden countries) to define the most relevant implementation strategies. Remote participation is encouraged to broaden inclusion and reduce costs, with careful attention given to workshop design, facilitation, internet access, and translation needs. This workshop may be combined with the Step Three workshop, potentially as an in-person or hybrid event.

Outputs:An implementation strategies overview, including detailed description of the seven Ws with their enabling factors and risks, identification of critical equity considerations, and listing of the expected benefits for users and providers.Documentation of the information gathering activities inclusive of the survey and interview questions, the sampling strategies, the country archetype definitions and supporting data, workshop notes, pre-reads, agenda, and participants.

### 2.3. Step Three—The Assessment of the Three Fits Within Each Implementation Strategy (Definition of the Use Cases)

Once the implementation strategies are defined, a user and product lens is adopted, switching the focus towards the practical use of the vaccine within those strategies. This implies looking again at the dimensions analysed in the first two steps in the context of the vaccinee–vaccinator interaction. Adopting a user and product lens requires a comprehensive portrayal of the relevant individuals involved in the vaccine’s use:The solution provider—the person who administers the vaccine, the vaccinator (the WHO), as shown in Box 3, Case Study 3—part 1.The solution user—the person who receives the vaccine, the vaccinee, and, where necessary, the caregiver involved (the WHOM).

In addition, the location where the vaccine is administered with all its characteristics and constraints (the WHERE) needs to be described. These aspects are mapped in Figure 6.

The users (WHO and WHOM) are redefined in this step. They are no longer the groups discussed in step two, but instead the individuals involved in the vaccine administration. Similarly, the location (the WHERE) is no longer the geographical setting but the physical location where the administration occurs.

Box 3Case Study 3—part 1: the definition of the “WHO” for MR vaccines with a microarray patch (MAP) presentation for a specific implementation strategy, i.e., the Solution–provider fit [25].Context: As part of a project to define the use cases and overall value proposition for an MR-MAP vaccine, a desk review was conducted, a user survey was administered to 111 immunisation stakeholders, and interviews were scheduled with 49 country representatives to identify the potential users capable of delivering the vaccine as part of different implementation strategies. Legal and regulatory requirements and constraints were assessed and documented. The findings were discussed and validated by the project steering group.
Example: MAPs are needle-free delivery devices that deliver dry vaccines just below the skin surface. The vaccine is delivered into the skin within seconds to minutes of their application. MR-MAPs are anticipated to be in a single dose presentation, eliminate the need for reconstitution, and have enhanced heat stability.
Considering the product attributes and modality of administration, different actors can potentially administer MR-MAPs as part of supplementary immunisation activity (SIA):
-Fully trained health workers (e.g., doctors, nurses, and registered pharmacists);Community workers with basic health training;Non-health personnel with no specific health training (e.g., teachers and community leaders);Caregivers or those performing self-administration.Programmatic and legal constraints were identified for the last two groups that were reflected in the subsequent use case definition.


The goal of this third step—the user–product analysis—is to develop an in-depth understanding of the real-life context in which users operate (e.g., family situation, financial situation, lifestyle, professional norms, habits, etc.), their feelings and beliefs, their thoughts about vaccines and vaccination, their acceptance of vaccines, and ultimately their needs that are to be satisfied by the vaccination programme either as vaccine recipients or as the individuals administering it. Personae can be developed to facilitate the investigation of differences in knowledge, attitudes, behaviours and practices (KABP) [26] that may influence their use of the vaccine. The term ‘Persona’ was developed by Swiss psychoanalyst Carl Jung and derived from the Latin word ‘persona’, which referred to the masks worn by Etruscan mimes. This concept has been adopted in design and market research to define fictional characters representing the different user types of a service or product. Applied to vaccines, the approach helps researchers capture (and visualise) users’ characteristics, particularly those more influential on vaccine use.

After mapping the WHERE, the WHO, and the WHOM, each “W” is assessed to determine its impact on the vaccine’s use. The result of this analysis is the selection of the dimensions that are most relevant for the definition of the vaccine use cases (Box 4, Case Study 3—part 2).

Box 4Case Study 3—part 2: prioritisation of the Ws for an MR MAPS vaccine [25].Context: As described in part 1.
Example: The dimensions that can impact how MR-MAP vaccines (the WHAT) are used and ultimately mapped:
-The delivery setting (the WHERE)—use in fixed posts with complete health services (such as a hospital or a health centre) compared to use in a context with limited availability or absence of health infrastructure (particularly of a cold chain or of trained health workers) was dependent on the setup of the immunisation system; this influenced the appropriate mix of delivery strategies (e.g., routine or campaign immunisation).The service providers (the WHO)—the type of service providers involved in the immunisation activities—health workers, community health workers, teachers, community leaders, caregivers, or persons performing self-administration—and their level of training and health knowledge can influence the acceptability and effectiveness of the MR-MAP administration.The targeted population (the WHOM)—the target age groups and their co-morbidities can trigger changes in the way a vaccine is used. In the case of MR vaccines, the focus on infants and young adults for campaigns reduced the variety of relevant use cases (e.g., the viability of self-administration or the ability to leverage specific professional or educational settings).The dimensions that most likely influence the use of MR-MAP vaccines were selected based on the results of the survey and interviews. The delivery setting (the “WHERE”), particularly the availability of a cold chain, and the service provider (the “WHO”), with a specific focus on the skill-level required for vaccine administration, were identified by the project steering group as having the biggest influence on how the vaccine was going to be used.


Once the most relevant dimensions identified, a mapping of the attributes of those dimensions follows. The prior analysis at the programmatic and policy level provides the foundation for this activity. These attributes are validated regarding their relevance in influencing vaccine use (Box 5—Case Study 4–part 1. This activity requires a forward-looking perspective that focuses on the “to be”, going beyond the “as is” view. Use cases that do not exist today may become viable in the future if new vaccines or improved versions of existing vaccines allow new uses.

Box 5Case Study 4—part 1: attributes of one of the shortlisted dimensions of typhoid conjugate vaccine (TCV) use cases [27].Context: As part of the definition of the value proposition for a TCV-MAP vaccine, a survey was administered to 155 respondents at the country, regional, and global level; four remote focus groups with 23 country participants were also organised. The findings were summarised by the working group and discussed with the steering group.
Example: The delivery location (the WHERE) was selected as a critical dimension given the MAP presentation. The following locations emerged as relevant:
-Public health facility (hospital, health centre, health post) with the delivery strategy fixed site with full cold chain;Private health facility (hospital, health centre, health post, private practice) with the delivery strategy fixed site with full cold chain;Private accredited pharmacy with the delivery strategy fixed site with full or reduced cold chain;Public setting with some health services (e.g., school, military barracks) with the delivery strategy outreach with reduced cold chain;Private setting with some health services (e.g., workplace, school, home) with the delivery strategy outreach with reduced cold chain;Public or private setting without health services (e.g., school, workplace, religious institution, other locations) with the delivery strategy mobile in absence of cold chain.

Following the definition of the most influential dimensions and their attributes, draft use cases are identified by looking at all potential combinations of the attributes along the shortlisted dimensions. This entails providing a precise description, a picture, of the different “uses” of a vaccine in the context of the immunisation programme’s efforts to achieve a disease control or elimination goal (Box 6, Case Study 5—part 1).

Box 6Case Study 5—part 1: seasonal influenza vaccine draft use cases [28].Context: As part of the definition of the full value proposition of seasonal influenza vaccines, use cases for influenza vaccines were defined. A desk review was conducted that resulted in the definition of the dimensions most likely to influence vaccine use. Review findings were discussed with the project steering group and led to a first drafting of the use cases.
Example: The key implementation strategy identified as relevant for seasonal influenza vaccination in the Northern Hemisphere was an annual, seasonal, time-limited campaign.The target populations (the WHOM) and delivery locations (the WHERE) were selected as the most relevant dimensions influencing the use of seasonal influenza vaccines.Five different target populations were identified based on the WHO recommendation: pregnant women, health workers, children, people with chronic conditions, and elderly people.
Three distinct delivery locations emerged as relevant when considering the countries using the vaccine after combining locations where the use of the vaccine was similar: health facilities (hospitals, health centres, health posts, private practices), accredited pharmacies (public or private), and settings with some or no health services (e.g., schools, military barracks, workplaces, religious institutions). Combining those dimensions and their attributes led to the definition of up to 15 potential use cases.

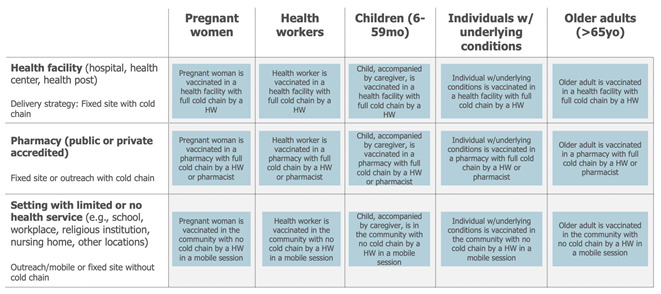


Lastly, the draft use cases are assessed to capture commonalities. A simplification of the use case framework usually occurs at this stage (Figure 7). This can be achieved by combining use cases whose practical differences in the use of the vaccine are minimal (e.g., because different target populations are receiving the vaccine in a similar way or because different locations share a similar mode of delivery).

The total population attributable to each use case can also be estimated to assess their size, an activity which supports prioritising the most relevant use cases (e.g., use cases with a very low total population may be consolidated with others). Also at this level, maintaining a forward-looking perspective that accounts for future epidemiological, demographic, or other dynamics potentially resulting in population changes is critical (Box 7, Case Study 5—part 2).

Box 7Case Study 5—part 2: use case refinement—sizing and prioritisation of the use cases of a seasonal influenza vaccine (continued from prior example) [28].Context: Following a first desk review (discussed in part 1), a survey administered to 287 experts, 48 country and global expert interviews, and a refined desk review were undertaken. These activities informed the sizing of use cases across different geographical contexts. Limitations were documented, and findings were validated by the project steering group. Based on the sizing results and the relevance of each use case, a subset of use cases was prioritised.
Example: The sizing of the use cases for a seasonal influenza vaccine focused on quantifying the maximum potential population that would use this vaccine in each of the identified use cases, independent of a time dimension and based on the current knowledge related to the delivery of seasonal influenza vaccines. The analysis assumed that these vaccines could be adopted contemporaneously by all countries and delivered to all priority target populations, irrespective of current seasonal influenza vaccine use and without any constraints related to policies (i.e., current national recommendations for influenza vaccines), financing (ability or willingness to pay), or programmatic (i.e., available cold chain or adult vaccination delivery approaches) or supply issues. The analysis resulted in the following assumptions about the different use case sizes—expressed as a proportion of each target population:

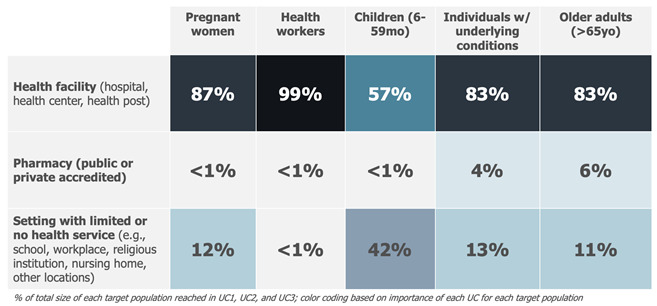
This result, combined with an analysis of the delivery modality, led to the exclusion of four smaller and less important use cases and to combining use cases for which the use of the vaccine was considered comparable. As a result, the following 9 use cases (reduced from 15) were defined for seasonal influenza vaccines:

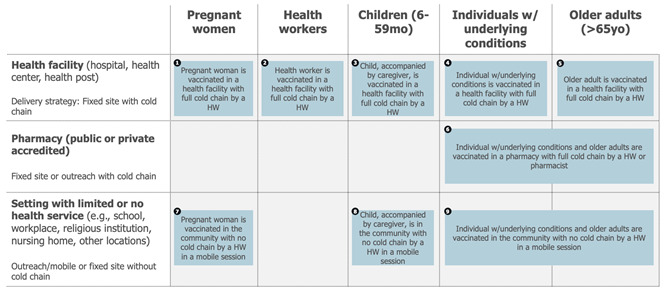


As mentioned in the programmatic and policy step of the analysis, contextual factors influence the implementation strategies and, consequently, the relative importance of the use cases. Reviewing the country archetypes can also help prioritise and further refine the use cases (Box 8, Case Study 4—part 2).

Box 8Case Study 4—part 2: use case refinement—country archetypes for a typhoid conjugate vaccine (TCV) MAP [27].Context: As described in part 1.
Example: Given the non-universal recommendations for TCV use and the heterogeneous burden within and between countries, country archetypes were developed for TCV-MAPs.
A standard set of characteristics were defined for these archetypes, which are also used for forecasting TCV-MAP demand. Eight potential parameters were reviewed with a focus on data availability:
-Income level as per World Bank classification (high income—HIC; upper-middle income—UMIC; low-middle income—LMIC; low income—LIC) with high data availability;Regional market characteristics (public, mixed private and public, etc.) with medium data availability;Typhoid incidence (high, medium, low) with high data availability;Level and type of typhoid Antimicrobial Resistance (AMR) severity (high, medium, low) with medium data availability;Level of typhoid surveillance (yes, no) with medium data availability;Level of access to water and sanitation (high, medium, low) with medium data availability;Existence of policy on typhoid control (yes, no) with low data availability;Historical use of typhoid vaccines (yes, no) with medium data availability.The first four parameters were chosen to define the country archetypes based on a review of the data in discussion with experts. This resulted in the definition of five country archetypes for TCV-MAPs:
-HICs or UMICs with low incidence and AMR;LMICs or LICs located in the WHO regions of the Americas (PAHO), Africa (AFR), Eastern Mediterranean (EMR), and Europe (EUR) with high typhoid incidence and AMR;LMICs or LICs located in PAHO, AFR, EMR, and EUR with medium typhoid incidence and AMR;LMICs or LICs located in PAHO, AFR, EMR, and EUR with low typhoid incidence and AMR;LMICs or LICs with significant private markets (located in the WHO region of South-Eastern Asia (SEAR) or the Western Pacific (WPR)) with high or medium typhoid incidence and AMR.

The final result of this step is a narrative description of the different “uses” of the vaccine for each implementation strategy. Continuing on to the example presented in step two, the narrative is expanded as follows (the bolded sentence describes one use case for the measles-rubella vaccine):

“*During an outbreak response managed by the NIP to a population aged 9 months to 15 years in countries in Central Asia with decentralised health systems and high vaccine hesitancy, with the goal of achieving measles elimination (the implementation strategy)*
**a lyophilised measles-rubella combination vaccine (the WHAT) is reconstituted and administered by a trained health worker (the WHO), to a 9-months old infant accompanied by the mother (the WHOM), in a remote health post without cold chain or storage capacity (the WHERE)**”

The use cases can be visualised in a matrix that synthesises the outcome of this third step and can serve as the basis for subsequent discussions and decisions. Figure 8 below presents the final output result of of the mapping of the use cases for a TCV-MAP vaccine. Some components of the work are described in the Part 1 and 2 of Case Study 4.

Upon completion of the three steps, a consolidated overview of the most likely implementation strategies and a detailed description of the use cases are generated. These outputs serve to inform the decision-making processes.

#### Methodology and Tools for the Definition of the Use Cases

Direct fieldwork with participation of the users is recommended for the identification of the use cases and for capturing user attitudes, preferences, and concerns. All field activities should be conducted by local researchers:If a prototype exists, direct observation at delivery points using structured checklists can be performed; alternative methods are remote walkthroughs, photo diaries, or moderated role-plays.If no prototype is available, product briefs can be used as the basis for interviews or focus group discussions with vaccinators and vaccinees/caregivers.

Based on the results of the stakeholder mapping of Step 2, adequate attention should be dedicated to capturing the perspectives of marginalised and hard-to-reach populations and of different health worker groups. If field work is not possible, remote interviews should be considered as an alternative.

Leveraging the results of the field work, the working group should formalise a Persona and Journey Mapping to describe provider and user goals, capabilities, and constraints, as well as end-to-end task flows and potential failure points.

A Use Case Definition and Prioritisation Workshop should be organised in regions where the disease is most relevant. Participants will include health workers, vaccinees and caregivers, policymakers, developers, regulators, funders, and implementing agencies. Adequate representation from high-burden countries (>40% of participants) must be ensured as well as representation of users. This workshop may be merged with Step Two’s workshop, and hybrid or virtual options can be adopted to broaden participation and reduce costs.

A final validation with users should be conducted by the country’s researchers through interviews with selected fieldwork participants aimed at gathering feedback on the workshop outputs. This may lead to final amendments to the use cases.

Outputs:A use case mapping, including personas, journey, and operational preconditions.A decision briefing describing the key product requirements and policy levers needed for the vaccine to be successful in its roll-out.Documentation of the field work and of the persona and journey mappings, as well as of the workshop with notes including pre-reads, agenda, participants, and main discussion points.

## 3. Discussion

The operational research method described in this paper aims at increasing clarity on a critical aspect of immunisation: how vaccines are being used. The approach positions the users—i.e., those that ultimately determine the success or failure of a programme [29,30]—at the centre of the research, while simultaneously emphasising the context in which immunisation is being delivered.

The proposed methodology leverages the design approach and emphasises the critical intersection of three fits: Solution–problem fit, Solution–user fit, and Solution–provider fit. The approach examines different factors (the seven Ws) that define the way in which the vaccine is used: first, it clarifies the public health goal; then it assesses the three fits at the programmatic–policy level by identifying the most impactful implementation strategies; finally, it conducts an assessment of the three fits at the user–product level to define vaccine use cases. This holistic approach is denominated “the Seven Ws” based on the initial letter of the labels of the seven dimensions analysed (WHY, WHAT, WITHIN, WHEN, WHERE, WHO and WHOM) and allows for a better alignment of vaccine product characteristics with both user and provider needs throughout the vaccine lifecycle.

The output of the seven W process provides both the description of the implementation strategies and of the potential use cases of the vaccine. In formulating those descriptions, the focus is forward-looking (the “to-be”) and outlining the full vaccine potential. Importantly, the sevenW process aims to stimulate critical discussions and decisions on product design to ensure that vaccines are better aligned with the users’ needs and the context in which they are used. The outcomes of those discussions should translate into product guidance documents that better incorporate users’ perspective into clinical development pathways that are more reflective of users’ needs, and, ultimately, into vaccines more suitable for use. The process is also valuable in itself, as it elicits a common understanding among stakeholders of the vaccine’s use cases.

Identifying these use cases should start in the early phases of clinical development when a number of product design, clinical development, and process design decisions can still be taken. However, the approach remains relevant across the entire life cycle of a vaccine, from early-stage clinical development to policy formulation and implementation. While the goal of maximising vaccine impact remains unchanged across the different stages, the focus of the user research and the level of information available varies.

In the early stages, with several product design decisions still pending, a higher degree of uncertainty is to be accepted. This uncertainty generally translates into a more extensive set of potential use cases. Achieving an initial level of clarity on the implementation strategies and use cases can inform clinical development design and investments, allowing for a more thoughtful and informed discussion about some of the potential use cases. In the specific, this clarity can also inform the design of the clinical trials ensuring that the necessary data are generated in the most relevant populations or in support of specific implementation strategies.

Later in the vaccine life cycle, use cases are more clearly defined and based on more solid evidence. At the same time, more decisions have already been made and product features defined. As a result, some of the potential use cases considered in the earlier phases may no longer be feasible. The focus of user research moves to optimising the programme design and to maximising users’ acceptability and ease of use based on given product characteristics. It can also inform the design of vaccination policies.

The adoption of this methodology can prove especially valuable for (a) vaccines targeting populations with a history of documented low vaccine coverage, (b) new presentations or formulations of existing vaccines that require changes in administration or in the supply chain, (c) vaccines intended for contexts with limited healthcare infrastructure or restricted resources, (d) situations where previous implementations have failed, and (e) innovations whose financial value propositions require multiple use cases to be commercially viable. In all those circumstances, the focus on user requirements can prove critical to ensure that the vaccine is suited to achieve its public health goals.

As with every methodology, limitations to its applicability exist and should be considered as researchers adopt this approach.

Firstly, some of the targeted populations may be difficult or impossible to reach (e.g., disadvantaged or discriminated groups, refugees, and hard-to-reach populations). In this case, important user voices may be missed, limiting the ability of capturing all relevant implementation strategies and use cases. The use of proxies that can provide at least some of the necessary insights should be explored.

Secondly, in the event of health emergencies, time is most likely not sufficient to perform the research by applying all of its steps. Furthermore, many users might not be accessible. Lastly and most importantly, the quick resolution of public health emergencies is the main priority, and this may lead to acceptance of product design features that could otherwise be considered problematic (e.g., mRNA COVID-19 vaccines requiring storage temperatures between −60 °C and −80 °C, which was very inconvenient for health workers and are largely unavailable in low-income settings). A more comprehensive approach can be adopted once an emergency has subsided to direct subsequent improvements in the product design. Pre-emptive research, such as the disease X approach adopted by CEPI, can also be considered.

Thirdly, the absence of a product prototype reduces the quality of insight collectable in field work. Discussing a theoretical, on-paper, product brief with final users may prove less effective than discussing a product prototype. Focusing on implementation strategies and on a high-level investigation of use cases should be considered in these circumstances. Once the product prototype is available, field work can be performed, seeking new insights that can lead to refinement of the use cases.

Lastly, on the process side, financial resources may be limited to perform all the steps of the research as described. The use of resource-sparing options should be considered. Alternatives are mentioned in the specific methodological paragraphs in each of the section above.

The application of this method requires careful consideration of its limitations and appropriate adaptations to ensure that the research outputs remain significant and relevant considering the specific context in which the study was conducted.

## 4. Conclusions

We hope that the proposed methodology and the examples provided can trigger discussions and draw attention to the critical importance of putting the users’ perspectives (the WHO and the WHOM) at the centre of vaccine development, policy, and implementation decisions, going beyond the traditional focus on the technical (the WHAT) and contextual aspects (the WHERE). In doing so, the direct involvement and input of the users at the policy, programme, and implementation levels is critical. Third parties should not represent vaccine users; on the contrary, users should be involved as active actors in the process of designing products and their implementation, with their voices heard throughout the decision-making chain. Ultimately, only by putting vaccine users at the centre and giving them a voice will we be able to progress towards achieving the IA2030 goals.

## Figures and Tables

**Figure 4 vaccines-13-01075-f004:**
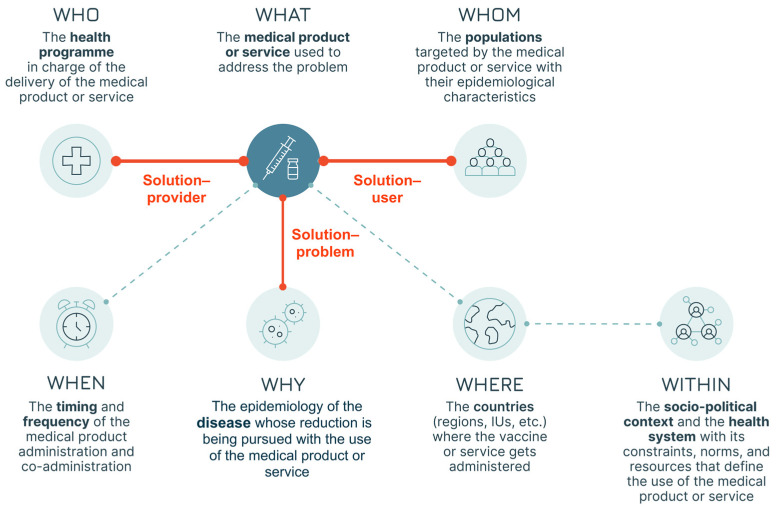
The seven elements of the policy and programmatic analysis.

**Figure 5 vaccines-13-01075-f005:**
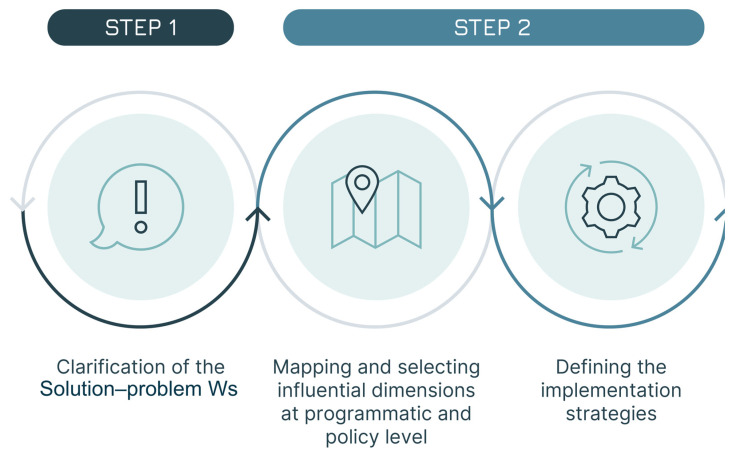
The first three steps of user research: the programmatic–policy analysis to define the implementation strategies.

**Figure 6 vaccines-13-01075-f006:**
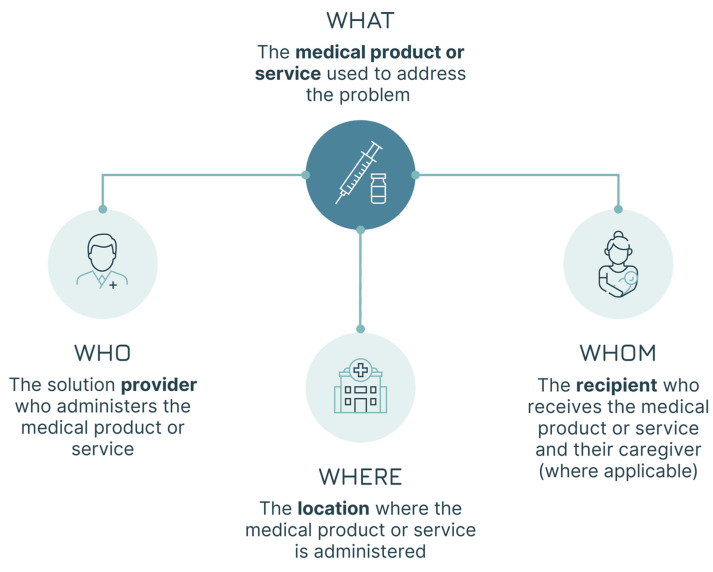
The four elements of the user–product analysis.

**Figure 7 vaccines-13-01075-f007:**
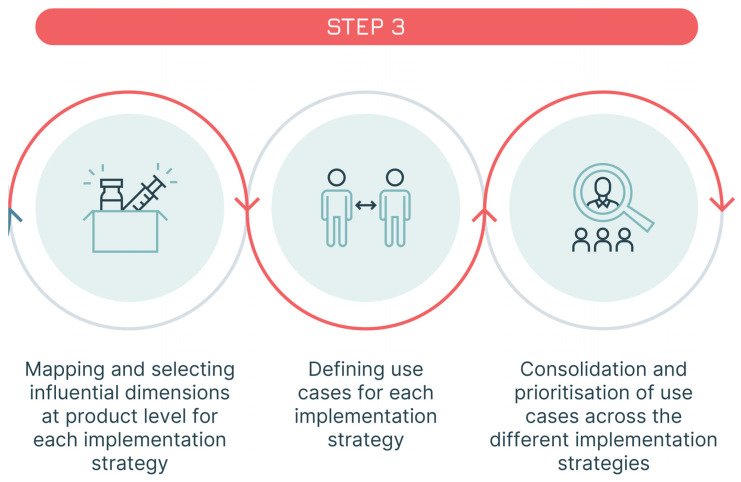
Step 3—identification of the use cases.

**Figure 8 vaccines-13-01075-f008:**
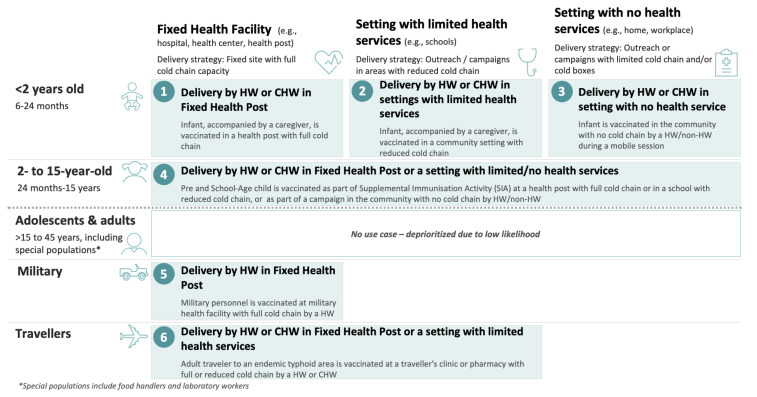
The 6 use cases for a TCV with MAPs presentation [27].

**Table 1 vaccines-13-01075-t001:** Key features of main guidance documents.

Document	When	Focus	Issued by
PPC	Very early stages of development (incl. preclinical) to guide research and innovation	High level characteristics that would maximise public health impacts, especially in L&MICs	WHO
ECVP	Early clinical development	Data and evidence required for policy recommendation in L&MICs	WHO
TPP	Throughout the clinical development cycle	Specific (minimally acceptable and preferred) characteristics for marketing authorisation	WHO, CEPI, CBER, …
PSPQ	Throughout the clinical development cycle	Product characteristics necessary for WHO PQ (required for UN procurement)	WHO

## Data Availability

Data relevant for the various case studies can be retrieved by accessing the referenced publications.

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
