# Peer review of "Improving the Suitability of Vaccine Design for Immunisation Programmes and Enhancing Vaccine Policy Quality Through User Research"

_vaccines, 2025, doi:10.3390/vaccines13101075_

Round 1
Reviewer 1 Report
Comments and Suggestions for Authors
Dear authors,
I have had the opportunity to review your manuscript title: "Improving the suitability of vaccine design for immunisation programs and enhancing vaccine policy quality through User
Research", this technical note presents an innovative methodological proposal called "the seven Ws" to systematically integrate user perspectives into vaccine development, policy formulation, and program implementation. The proposed methodology addresses a critical gap identified by the authors: that current guidance documents primarily capture technical aspects of the product, with the perspectives of users (healthcare workers, vaccine recipients, and caregivers) frequently captured only ex-post to validate technical aspects of vaccines. In this case, your approach is structured in three steps that systematically assess solution-problem, solution-user, and solution-provider fits, with the aim of facilitating the achievement of the goals of the 2030 Immunization Agenda.
Main Strengths:
The methodological proposal responds to a real and well-documented need in the field of vaccine development. These authors present compelling examples of suboptimal design decisions, such as rotavirus vaccines with large packaging volumes that strain the cold chain, pneumococcal conjugate vaccines developed by selecting serogroups most prevalent in North America and Europe, and COVID-19 mRNA vaccines that require ultra-cold cold chain temperatures.
The three-fits conceptual framework (solution-problem, solution-user, solution-provider) is firmly grounded in established design principles and provides a comprehensive analytical structure. The approach ideally extends to the entire lifecycle of a vaccine, from its design and development to all aspects of service delivery, enabling early insights into vaccine deployment strategies and user needs to help anticipate potential policy recommendations.
The case studies presented (Pansarbecovirus, tuberculosis, seasonal influenza, microarray patching), effectively demonstrate the applicability of the proposed approach in diverse epidemiological and programmatic contexts, illustrating the methodology's versatility and transferability.
Sections for Consideration:
- Explanation of Methodological Scope: while the proposal is conceptually accurate, it would be valuable if the authors briefly elaborated on the specific criteria for determining when the proposed methodology would be most beneficial versus traditional vaccine development approaches.
- Integration with Current Frameworks: Although existing guidance documents (PPC, TPP, ECVP, PSPQ) are mentioned, is necessary a more detailed discussion of how this new methodology would specifically integrate with established regulatory and decision-making processes would be helpful.
- Practical Implementation Considerations: The authors appropriately recognize resource constraints, but a brief discussion of prioritization strategies required when applying the full methodology is not feasible would strengthen the proposal presented in this paper.
Specific Comments
Methodological Framework:
- The adapted definition of "use case" from software development as "a specific situation where a health product or service is or can be used to achieve a defined health objective" is appropriate and facilitates understanding of the approach.
- The logical progression of the three steps (general solution-problem fit → implementation strategies → specific use cases) is methodologically sound.
Case Studies:
- Case Study 1 (defining the WHY for a Pansarbecovirus vaccine) effectively demonstrates the complexity of defining public health objectives in different epidemiological contexts.
- Case Study 2 (defining the WHO for TB vaccines) appropriately illustrates the multidimensionality of identifying target populations.
Tools and Methods:
- The methodological tools described (literature review, facilitated workshops, direct field observation) are appropriate, and the authors adequately acknowledge alternatives when direct observation is not feasible.
SPECIFIC REVISIONS REQUIRED IN THE MANUSCRIPT
- Applicability Criteria (lines 605-620):
Authors should incorporate a new subsection entitled 'Methodology Applicability Criteria' that explicitly specifies when this methodology would be most beneficial versus traditional approaches. This section should include:
- Vaccines targeting populations with a history of documented low vaccine coverage
- New presentations or formulations of existing vaccines that require changes in the administration chain
- Vaccines intended for contexts with limited healthcare infrastructure or restricted resources
- Situations where previous implementations have failed according to published evidence
- Integration with Regulatory Frameworks (Expansion required on lines 606-611):
The authors should expand this section by including:
- Clinical phase timeline: Specify the phases of clinical development (Phase I, II, III, Post-marketing) each stage of the analysis would be more critical
- Regulatory Integration: Describe how the results would specifically inform EMA/FDA decisions and their incorporation into TPP (Target Product Profile) and PPC (Preferred Product Characteristics) documents
- Implementation Timeline: Specific timeline for when to apply each of the "seven Ws" in relation to established regulatory milestones
- Limitations and contexts of lesser applicability (new mandatory section in Discussion):
Include a new subsection titled 'Methodological Limitations and Contexts of Restricted Applicability' that should address:
- Human and economic resource limitations for a full implementation
- Contexts where traditional approaches might be more efficient Health emergency situations where the methodology might be impractical
- Restrictions on specific populations or contexts where obtaining user insights is limited.
- SUCCESS INDICATORS (Addition required before Conclusions)
Authors must include a subsection on 'Methodology Evaluation Indicators' that proposes specific metrics to assess implementation success, such as:
- Process indicators (number of user groups consulted, completeness of the seven Ws)
- Outcome indicators (improvement in acceptability, feasibility of implementation)
- Comparative metrics with traditional vaccine development approaches
- EXPLICIT METHODOLOGICAL STATEMENT (Modification required in Abstract/Introduction)
Mandatory addition:
Explicitly include in the abstract and in the first paragraph of the introduction the statement: 'This article presents a methodological proposal to systematically integrate user perspectives in vaccine development, policy, and implementation, contributing to the conceptual framework of the field without constituting an empirical validation study
Assessment of Specific Aspects of your article:
Regarding to Figures and Tables: The figures are clear and effectively communicate the core concepts. Figure 1 illustrates this methodology particularly well.
The concept of the three adjustments, and Figure 4 appropriately presents the complexity of the elements of programmatic and policy analysis.
Structure and Presentation: The manuscript is well-structured, with a logical progression from problem identification to proposed solutions and illustrative examples.
Overall Recommendation
The methodological proposal presented represents a valuable and timely contribution to the field of vaccine development and implementation. The authors are clear and explicit that their objective is to "encourage the discussion and draw attention to the critical importance of placing user perspectives at the center of vaccine development, policy, and implementation decisions" and emphasize that "only by placing users at the center and giving them a voice can we make progress toward achieving the AI2030 goals."
The proposed methodology addresses a recognized gap in current approaches and provides a well design systematic framework for integrating user perspectives. The case studies effectively demonstrate the practical applicability of the approach in diverse contexts.
In my opinion the Final Recommendation for this article is: Accept after Minor Revisions
All the best
Author Response
Dear Reviewer One,
Thank you very much for your feedback that helped us greatly improve the clarity and the rigor of our work. We have worked to address all your recommended actions, and we are providing an overview of the implemented changes to facilitate the identification of these changes in the revised version of the paper that we have just submitted.
- Explanation of Methodological Scope: while the proposal is conceptually accurate, it would be valuable if the authors briefly elaborated on the specific criteria for determining when the proposed methodology would be most beneficial versus traditional vaccine development approaches.
The proposed methodology is meant to integrate existing approaches by emphasizing the users’ perspective and ensuring its consideration when guidance documents or development, policy, or implementation decisions are taken. We have added clarification in line 110 and following.
- Integration with Current Frameworks: Although existing guidance documents (PPC, TPP, ECVP, PSPQ) are mentioned, is necessary a more detailed discussion of how this new methodology would specifically integrate with established regulatory and decision-making processes would be helpful.
We have expanded on the topic in line 182 and following.
- Practical Implementation Considerations: The authors appropriately recognize resource constraints, but a brief discussion of prioritization strategies required when applying the full methodology is not feasible would strengthen the proposal presented in this paper.
Establishing resource-sparing approaches and methods is critical, especially in those times of financial constraints for countries, researchers and global organisations. We have expanded on this aspect in the methods and tools paragraphs for each of the three steps (paragraphs 2.1.1, 2.2.1, 2.3.1) clarifying how the research can be performed extensively using virtual and remote approaches and leveraging to the maximum extent local researchers. While their involvement should be primarily driven by ownership and country knowledge, local researchers' participation can greatly reduce the need for travel and related costs. We also explicitly mentioned the resource constraints as one of the potential limitations of the approach in line 819 and following.
- Applicability Criteria - Authors should incorporate a new subsection entitled 'Methodology Applicability Criteria' that explicitly specifies when this methodology would be most beneficial versus traditional approaches.
As mentioned above, we do not see this methodology as an alternative to the current approaches. We are now discussing specific examples where the method can prove especially valuable in line 787 and following.
- Integration with Regulatory Frameworks
- Clinical phase timeline: Specify the phases of clinical development (Phase I, II, III, Post-marketing) each stage of the analysis would be more critical
- Regulatory Integration: Describe how the results would specifically inform EMA/FDA decisions and their incorporation into TPP (Target Product Profile) and PPC (Preferred Product Characteristics) documents
- Implementation Timeline: Specific timeline for when to apply each of the "seven Ws" in relation to established regulatory milestones
Discussion has been added in line 180 and following and expanded on in line 766 and following concerning the timing in the development cycle when the approach can best contribute.
With regard to the regulatory aspects, as this process provides relevant input to clinical development, it also does for the regulatory strategy. At the same time, some of the product characteristics that may emerge from the use case work may not be viable from a regulatory standpoint. We have clarified in line 193 and following how a feedback loop needs to be activated as part of the downstream process whereby the product design elements emerging from the insights collected need to be viable also in terms of regulatory strategy.
- Include a new subsection titled 'Methodological Limitations and Contexts of Restricted Applicability' that should address:
- Human and economic resource limitations for a full implementation
- Contexts where traditional approaches might be more efficient Health emergency situations where the methodology might be impractical
- Restrictions on specific populations or contexts where obtaining user insights is limited
Limitations of the method and constraints to its applicability are now discussed in line 796 and following as part of the Discussion. Specifically, the following instances have been mentioned: resource constraints, availability (or lack thereof) of a product prototype, feasibility of this approach in case of health emergencies, as well as in the ability of involving or reaching certain populations. For each of them potential alternative approaches are suggested.
- Authors must include a subsection on 'Methodology Evaluation Indicators' that proposes specific metrics to assess implementation success, such as:
- Process indicators (number of user groups consulted, completeness of the seven Ws)
- Outcome indicators (improvement in acceptability, feasibility of implementation)
- Comparative metrics with traditional vaccine development approaches
Suggested process indicators have been added in the three method and tools subparagraphs of the three steps (paragraphs 2.1.1, 2.2.1, 2.3.1); outputs of the activities are now listed. With reference to outcome indicators, we have not included any at this stage. In view of the novelty of the approach, we would prefer to limit our outcome claim to the conservative generic reference to the contribution to the IA2030 outcomes currently included in the text. We are happy to consider including more specific references (e.g. acceptability, coverage improvements and reduction of wastage), if this is deemed appropriate.
- Mandatory addition: explicitly include in the abstract and in the first paragraph of the introduction the statement: 'This article presents a methodological proposal to systematically integrate user perspectives in vaccine development, policy, and implementation, contributing to the conceptual framework of the field without constituting an empirical validation study
The recommended text has been added in the conclusion section of the abstract and in line 178 and following.
We want to thank you again for the important and stimulating inputs. We sincerely appreciated them and found very useful to improve the quality of the paper and for clarifying important aspects of the method that were not explicit in the first submission. We hope that the implemented amendments address your concerns and remain open for continued discussion on the topic.
Sincerely, on behalf of the authors
Stefano Malvolti
Reviewer 2 Report
Comments and Suggestions for Authors
The technical note by Malvolti et al. is well-organized and clearly outlines the central issue. The overall flow of writing is very smooth and honestly I enjoyed reading it. From identifying gaps in current practice to proposing a framework. The use of case studies will really help readers. There are some minor concerns which needs to be addressed and some suggestions for better clarity.
I believe authors should define all seven “W” factors earlier in the manuscript. Currently, the seven W’s (WHY, WHAT, WHERE, WITHIN, WHEN, WHO, WHOM) are introduced in pieces. A brief explanation altogether before you detail at different places will be a good idea and easily understandable.
There is also one potential confusion i.e., “the WHO” and “the WHOM”. Because “WHO” is a well-known acronym for the World Health Organization, some readers might misinterpret “the WHO” in this context. Though it is somehow understandable, but a brief clarification would be helpful. OR you can use some other term.
In methodology section, I think if you add one paragraph to briefly mention any limitations or challenges in applying this approach and how to manage such challenges could enhance the reproducibility.
Implementing these suggestions would polish the manuscript further and will be an excellent contribution to the literature on vaccine policy and product development.
Author Response
Dear Reviewer Two,
Thank you very much for your feedback that helped us greatly improve the clarity and the rigor of our work. We have worked to address all your recommended actions, and we are providing an overview of the implemented changes to facilitate the identification of them in the revised version of the paper that we have just submitted.
- I believe authors should define all seven “W” factors earlier in the manuscript. Currently, the seven W’s (WHY, WHAT, WHERE, WITHIN, WHEN, WHO, WHOM) are introduced in pieces. A brief explanation altogether before you detail at different places will be a good idea and easily understandable.
The 7 Ws are now introduced altogether in a succinct way before the description of the three phases in line 207 and following.
- There is also one potential confusion i.e., “the WHO” and “the WHOM”. Because “WHO” is a well-known acronym for the World Health Organization, some readers might misinterpret “the WHO” in this context. Though it is somehow understandable, but a brief clarification would be helpful. OR you can use some other term.
Clarification is provided in lines 217–218 to avoid misunderstanding with reference to the use of WHO vs. W.H.O.
- In methodology section, I think if you add one paragraph to briefly mention any limitations or challenges in applying this approach and how to manage such challenges could enhance the reproducibility.
Limitations of the method and constraints to its applicability are now discussed in line 796 and following as part of the Discussion. Specifically, the following instances have been mentioned: Impact of resource constraints, of the availability (or lack thereof) of a product prototype, of the feasibility of this approach in case of health emergencies, as well as in the ability of involving or reaching certain populations. For each of them potential alternative approaches are suggested.
We want to thank you again for the important and stimulating inputs. We sincerely appreciated them and found very useful to improve the quality of the paper and for clarifying important aspects of the method that were not explicit in the first submission. We hope that the implemented amendments address your concerns and remain open for continued discussion on the topic.
Sincerely, on behalf of the authors
Stefano Malvolti
Reviewer 3 Report
Comments and Suggestions for Authors
This paper proposes a user-centered design methodology—the “seven Ws” framework—to improve vaccine design and policy decisions by integrating the perspectives of users (vaccinees, caregivers, vaccinators) and implementers (health systems, governments). The authors argue that current global guidance (e.g., WHO PPCs, TPPs, ECVP) focuses narrowly on technical attributes (efficacy, dosing, cold chain) and neglects user needs (acceptability, convenience, cultural fit). They advocate for early and continuous user research to ensure vaccines are equitable, accessible, and effective, particularly in low- and middle-income countries (L&MICs). The paper includes case studies on vaccines like pan-sarbecovirus, TB, MR-MAPs, TCV, and influenza to illustrate the methodology.
While the paper presents a valuable and timely contribution, it suffers from significant methodological, conceptual, and practical weaknesses that undermine its suitability for publication in its current form. Below are the key issues, exemplified point-by-point:
1# The framework is purely descriptive and lacks quantitative or comparative validation. Example: Case Study 6 (influenza use cases) relies on expert opinion rather than field data or modeling. No evidence is provided that the seven Ws outperform existing tools like TPPs.
2# Use cases are speculative (e.g., self-administration of MR-MAPs by caregivers), with no feasibility testing. Example: The claim that non-health workers (teachers, community leaders) could administer MAPs is unsubstantiated—no regulatory or safety evidence is cited.
3# The paper conflates “user research” with stakeholder workshops/interviews, ignoring power dynamics (e.g., donors vs. marginalized users). Such as, no discussion of how to include hard-to-reach populations (e.g., nomadic groups, undocumented migrants).
4# The framework overemphasizes design while ignoring systemic constraints (e.g., financing, corruption, political instability). The paper critiques mRNA vaccines for UCC requirements but offers no solutions for infrastructure gaps in L&MICs.
5# The term “user” shifts ambiguously between individuals (vaccinees) and institutions (NIPs, donors). In Case Study 2 (TB vaccine), “WHOM” includes prisoners and miners, but their agency in design decisions is unexplored.
6# The design approach is imported from HIC contexts (e.g., MAPs, self-administration), which may not align with L&MIC realities. Self-administration assumes literacy and health literacy, which are low in many L&MIC settings.
7# The paper claims to center users but positions them as passive informants rather than co-designers. Users are “consulted” in workshops, but no mechanism ensures their feedback influences decisions.
8# The methodology requires extensive fieldwork (direct observation, workshops), which is cost-prohibitive for most L&MICs. Provide lightweight alternatives for resource-limited settings.
The paper fails to demonstrate that the seven Ws framework is superior to existing tools or feasible for L&MICs. Major revision is guaranteed. Without these revisions, the paper risks being an academic exercise rather than a practical tool for vaccine equity.
Author Response
Dear Reviewer Three,
Thank you very much for your feedback that helped us greatly improve the clarity and the rigor of our work as well as its applicability. We have worked extensively to address all your recommended actions across all sections of the document. We are here providing an overview of the implemented changes to facilitate the identification of them in the revised version of the paper that we have just submitted.
1# The framework is purely descriptive and lacks quantitative or comparative validation. Example: Case Study 6 (influenza use cases) relies on expert opinion rather than field data or modeling. No evidence is provided that the seven Ws outperform existing tools like TPPs.
Our proposed approach is not meant to replace existing tools or approaches but rather to enrich them by ensuring that the users' needs are discussed at the right moment and in a systematic fashion to inform more holistically the products and program design. The proposed method, in combination with others, can support the definition of the ECVP as well as a critical review of the TPP. This has been made explicit in the Introduction (line 110 and following,180 and following ) and in the Discussion (line 787 and following).
2# Use cases are speculative (e.g., self-administration of MR-MAPs by caregivers), with no feasibility testing. Example: The claim that non-health workers (teachers, community leaders) could administer MAPs is unsubstantiated—no regulatory or safety evidence is cited
The five case studies are provided to illustrate how to implement the method for specific steps – all but one (the first one that is leveraging unpublished work) draw from published research. We have added a context paragraph at the start of each of the case study (line 267 and following, 354 and following, 476 and following, 552 and following, 578 and following, 614 and following) to provide an overview of the specific context behind each of the examples without having to access the full papers.
In the specific, for Influenza, the research was just one part of the broader WHO’s full value assessment of seasonal influenza vaccines (https://www.who.int/teams/immunization-vaccines-and-biologicals/immunization-analysis-and-insights/vaccine-impact-value/full-value-of-improved-influenza-vaccine-assessment-(fviva)), and aimed at generating a first overview of the targeted populations with some idea on the potential use cases. This output served as input for other modelling work. The papers on the modelling is currently in finalisation as is the one that describe the whole full value assessment.
For MAPs, interviews and surveys were performed that included several country informants which led to the definition of the uses case this is now mentioned in the context paragraphs mentioned above. As part of this process the programmatic and legal constraints that may apply to the non-health workers delivering the map were recorded and are explicitly mentioned in the research. Recognising the importance of this point, we explicitly mentioned this important outcome in the specific case study (lines 496-6)
For what concern the regulatory aspects, as this processs provides relevant input to clinical development, it also does for the regulatory strategy. At the same time, some of the product characteristics that may emerge from the use case work may not be viable from a regulatory standpoint. We have clarified in line 184 and following how a feedback loop needs to be activated as part of the downstream processs whereby the product design elements emerging from the insights collected need to be viable also in terms of regulatory strategy.
3# The paper conflates “user research” with stakeholder workshops/interviews, ignoring power dynamics (e.g., donors vs. marginalized users). Such as, no discussion of how to include hard-to-reach populations (e.g., nomadic groups, undocumented migrants).
We have now made explicit that a stakeholders mapping is foreseen at start, and that should raise awareness of power dynamics and inform the subsequent insight gathering steps (e.g., workshops/interviews/surveys/field research). The importance of appropriate sampling and design strategies that allow to account for the impact of those power dynamics is now made explicit in line 418 and following. Similarly, the need to capture the perspective all of all groups and communities is noted for the field work in line 712 and following. Lastly, the critical importance of capturing first-hand the voices of the users, including those often underrepresented is also reiterated in the conclusion (line 832 and following).
4# The framework overemphasizes design while ignoring systemic constraints (e.g., financing, corruption, political instability). The paper critiques mRNA vaccines for UCC requirements but offers no solutions for infrastructure gaps in L&MICs.
On the systemic constraints that affect health systems and the immunization systems, those mentioned as one of the goals of the analysis of the WITHIN (line 336 and following). We are not expanding on the topic to avoid extending even further the length of the paper, but, we can include explicit examples if this can help further improving the clarity.
On mRNA vaccine, there was no intention of critiquing. There is full acknowledgement that the timing did not allow for any extensive discussion on product characteristics and the public health issue so pressing that making a vaccine available was the single most important goal. Nonetheless, the implementation issues linked to the UCC are documented and cannot be ignored. At the same time, we acknowledge that the proposed method is not appropriate in the event of a public health emergency. We have clarified this in line 805 and following. In that paragraph a staged approach is suggested that may allow for a subsequent assessment of the use cases that can inform second generation vaccines.
5# The term “user” shifts ambiguously between individuals (vaccinees) and institutions (NIPs, donors). In Case Study 2 (TB vaccine), “WHOM” includes prisoners and miners, but their agency in design decisions is unexplored
We have expanded on the fact that the processs iterates on the WHOM, WHO and WHERE, one first time at user groups / population level to inform the implementation strategy definition – line 342 and following - and on second time at the individual user level for the use case definition – line 456 and following.
On the case study on the TB vaccine, miners and prisoners were included as potential target populations as per the input provided by 27 country informants across 10 countries. We have clarified the context in line 354 and following. The need to explore programmatic implications via the established channels (e.g. SAGE, RITAG and NITAG) was addressed in the study. This aspect is not mentioned in this case study since the example is introduced with the goal of illustrating only the step of target populations selection.
6# The design approach is imported from HIC contexts (e.g., MAPs, self-administration), which may not align with L&MIC realities. Self-administration assumes literacy and health literacy, which are low in many L&MIC settings
The method is not only for L&MICs, it applies to all settings; if HICs are epidemiologically relevant, use cases relevant for those countries are also in scope for our proposed approach. This can be of special relevance in dual market contexts and full value proposition assessment . This aspect has been clarified in line 791 and following.
7# The paper claims to center users but positions them as passive informants rather than co-designers
Our work is driven by the belief that capturing users’ needs and perspective is a necessity if we want vaccines, vaccine policies and immunization programs to improve. Pragmatically, and recognising the practical and cultural barriers that often hinder a direct involvement of the users, we have defined an approach that can work both with the users in the position of informants as well as co-designer. At the same time, we fully agree on the need for creating the conditions for a more active engagement and to start working removing the barriers that prevent such an involvement. We have explicitly included reference to the need for a more active involvement in line 422 and following, line 703 and following and in the conclusions in line 832 and following.
8# The methodology requires extensive fieldwork (direct observation, workshops), which is cost-prohibitive for most L&MICs. Provide lightweight alternatives for resource-limited settings
Limitations of the method and constraints to its applicability are now discussed in line 796 and following as part of the Discussion. In the specific the following instances have been mentioned: resource constraints, availability (or lack thereof) of a product prototype, feasibility of this approach in case of health emergencies, as well as in the ability of involving / reaching certain populations. For each of them potential alternative approaches are suggested.
The paper fails to demonstrate that the seven Ws framework is superior to existing tools or feasible for L&MICs.
As indicated in the conclusion, our intention is for contributing to the discussion on these important topics not to set a new standard. With this goal in mind, the proposed approach is not intended to replace existing ones but to complement them by providing a tool for more systematically capturing and discuss users’ needs and perspectives.
Potential constraints that may hinder its applicability for L&MICs have been discussed in the method parts and resource-sparing alternatives indicated to facilitate its adoption.
We want to thank you again for the important and stimulating inputs. We sincerely appreciated them and found very useful to improve the quality of the paper and for clarifying important aspects of the method that were not explicit in the first submission. We hope that the implemented amendments address your concerns and remain open for continued discussion on the topic.
Sincerely, on behalf of the authors
Stefano Malvolti
Round 2
Reviewer 3 Report
Comments and Suggestions for Authors
Most of the comments have been addressed.